

# Sucrose rinse modulates the salivary behavior of carbonic anhydrase VI and its buffering capacity: a longitudinal study in 4 to 6.5-year-old children

Thayse Rodrigues de Souza[1], Bruna Raquel Zancope[1], Emerson Tavares de Sousa[1], Thais Manzano Parisotto[2], Marcelo Rocha Marques[3] and Marinês Nobre dos Santos[1]

[1] Department of Health Sciences and Pediatric Dentistry, Piracicaba Dental School, Universidade de Campinas, Piracicaba, São Paulo, Brazil
[2] Department of Microbiology and Molecular Biology, São Francisco University Dental School, Bragança Paulista, São Paulo, Brazil
[3] Department of Morphology, Piracicaba Dental School, Universidade de Campinas, Piracicaba, São Paulo, Brazil

Corresponding author
Marinês Nobre dos Santos, mnobre@unicamp.br

## ABSTRACT

**Background:** Carbonic anhydrase VI (CA VI) is crucial in regulating oral pH and predicting susceptibility to dental caries. The hypothesis posits that caries activity may alter the CA VI function, diminishing its capacity to regulate pH effectively and potentially exacerbating cariogenic challenges. This 1-year cohort study sought to investigate the enzymatic activity of salivary CA VI and buffering capacity following a 20% sucrose rinse in 4 to 6.5-year-old children.

**Method:** This research involved 46 volunteers categorized into three groups based on their caries status after follow-up: caries-free ($C_{Fee}$), arrested caries ($C_{Arrested}$), and caries active ($C_{Active}$). Children underwent visible biofilm examination and saliva collection for salivary flow rate, buffering capacity, and CA VI analyses before and after a 20% sucrose rinse.

**Results:** A reduction in the buffering capacity was observed after sucrose rinse in all groups. The CA VI activity decreased significantly in $C_{Fee}$ and $C_{Arrested}$ groups after sucrose rinse, although it did not change in the $C_{Active}$ group. An improvement in the buffering capacity and salivary flow rate was found at follow-up when compared with the baseline. After 1-year follow-up, buffering capacity and salivary flow rate increased in all groups, whilst the CA VI activity reduced only in $C_{Free}$ and $C_{Arrested}$ children.

**Conclusion:** Sucrose rinse universally reduces the salivary buffering capacity, while caries activity may disrupt CA VI activity response during a cariogenic challenge. After a year, increased salivary flow enhances buffering capacity but not CA VI activity in caries-active children.

## INTRODUCTION

Saliva creates a protection against dental caries due to the natural interplay of organic and inorganic components that rules its physicochemical properties toward a healthy state (*Dawes, 2003*). Among these properties, buffering capacity is widely recognized as a caries-risk predictor since it demonstrates the individual ability to neutralize the tooth-biofilm interface and establish the ion-supersaturated state necessary for tooth surface remineralization (*Leone & Oppenheim, 2001*; *Tenovuo, 1997*).

The bicarbonate buffer system plays a crucial role in the acid-base homeostatic mechanism of stimulated saliva (*Pitts et al., 2017*), which occurs through the two-step reaction $CO_2 + H_2O \leftrightarrow H_2CO_3 \leftrightarrow HCO_3^- + H^+$. Due to its slow nature, carbonic anhydrase VI catalyzes the $CO_2$ hydration/carbonic acid ($H_2CO_3$) dehydration reaction to help achieve a pH close to physiological levels. The second reaction, involving the natural dissociation or ionization of $H_2CO_3$, it is more spontaneous (*Kivelä et al., 1999a*; *Breton, 2001*; *Occhipinti & Boron, 2019*). This enzyme can be found in saliva due to the serous-acinar secretion from human parotid and submandibular glands (*Parkkila et al., 1990*).

Recent studies suggest that dental caries can notably impact the CA VI isoform. This enzyme has been implicated in a high propensity to form dental biofilms dominated by aciduric and acidogenic species, increasing the risk of dental caries (*Esberg et al., 2019*). Additionally, both the concentration (*Szabó, 1974*; *Kivelä et al., 1999b*; *Oztürk et al., 2008*; *Picco et al., 2017, 2019*) and the enzymatic activity (*Frasseto et al., 2012*; *Cardoso et al., 2017*; *Borghi et al., 2017*; *Picco et al., 2017, 2019*; *Lima-Holanda et al., 2021*; *de Sousa et al., 2021*; *de-Sousa, Lima-Holanda & Nobre-Dos-Santos, 2021*) of CA VI in whole saliva have been correlated with caries experience. However, the reciprocal significance between them has not been fully understood.

Saliva interacts with the tooth surface, biofilm, and mucosa, undergoing changes that can reflect the characteristics of the host and the oral environment (*Marsh, Head & Devine, 2015*; *Proctor, 2016*). Therefore, dietary sugar metabolism, microorganisms, and caries activity can significantly induce saliva modifications (*Lenander-Lumikari & Loimaranta, 2000*; *Lips et al., 2017*; *Pitts et al., 2017*). According to prior research, children with caries exhibit notably higher CA VI activity in both saliva and biofilm compared to caries-free children (*Frasseto et al., 2012*; *Picco et al., 2017, 2019*; *de Sousa et al., 2021*). This outcome can be attributed to an enzymatic adaptation driven by repeated pH fluctuations after frequent cariogenic challenges. Noteworthy, the imbalance that occurs in caries microbiome dysbiosis can cause a biochemical shift in the oral environment (*Marsh, 2018*) and possibly change the molecular structure of biological molecules (*Zeng, 2011*; *Belda-Ferre et al., 2015*; *Buonanno et al., 2018*). This process may be integrated to make individuals more susceptible to developing caries lesions (*de-Sousa, Lima-Holanda & Nobre-Dos-Santos, 2021*).

Interestingly, salivary buffering capacity was negatively correlated with CA VI activity (*Picco et al., 2022*). These findings have raised the hypothesis that the increase in CA VI activity in the caries-affected group may not be associated with an enhanced buffering

effect. Evaluating changes in salivary function following a transient cariogenic challenge can provide valuable insights into understanding the deleterious effects of sucrose in individuals at high risk of caries. Further research is needed to comprehensively understand the interplay between CA VI and dental caries physiopathology, including the underlying mechanisms that may impact saliva's buffering capacity.

Given this background, this cohort study aimed to investigate the behavior of CA VI activity and buffering capacity in 4- to 6.5-year-old children after a 20% sucrose rinse. The 20% sucrose concentration simulates a high cariogenic challenge found in soft drinks, cakes, and biscuits. Also, some researchers have demonstrated that the physiology of salivary function can be affected by exposure to 20% sucrose rinse (*Frasseto et al., 2012*; *de-Sousa, Lima-Holanda & Nobre-Dos-Santos, 2020*, *2021*; *Lima-Holanda et al., 2021*).

# MATERIALS AND METHODS

## Ethical considerations

This study was approved by the Research Ethics Committee of the Piracicaba Dental School, Universidade Estadual de Campinas, under protocol No. 0142012. Volunteers' parents signed an informed consent form after being thoroughly instructed about the study procedures. Children with dental treatment needs were referred to comprehensive dental care at the Piracicaba Dental School of the Universidade Estadual de Campinas.

## Subjects

Two independent and normally distributed populations were used for the sample size calculation considering a two-sided test power provided by Gpower 3.1 program. A study that evaluated salivary CA VI activity in preschoolers was chosen for the sample estimation (*de Sousa et al., 2021*), where the mean (standard deviations) values were 10.03 (11.62) and 22.15 (17.81) for caries-free and caries-affected children, respectively. The input parameters were 0.05 $\alpha$-value, 0.10 $\beta$-value, 1/1 allocation rate, and 0.95 confidence interval. The calculated number was 35 children for each group. However, this number was increased to 41, considering possible losses to follow-up.

A probabilistic single-clustering method determined the sampling procedure. The Municipal Department of Health of Piracicaba-SP randomly selected two public nurseries, where children were subjected to an eligibility analysis. The exclusion criteria were systemic diseases, disabilities, use of antibiotics or nervous system drugs, severe fluorosis, dental hypoplasia, and children who refused to comply with any study phase.

## Study design

Figure 1 shows a diagram of the experimental design. In total, 300 potentially eligible children were invited to participate, but only 35% were examined for eligibility and submitted to clinical examination to diagnose dental caries. At baseline, 104 children of both sexes, with a gender ratio of 1 girl to 0.96 boys, aged between 48 and 78 months, were divided into two groups:

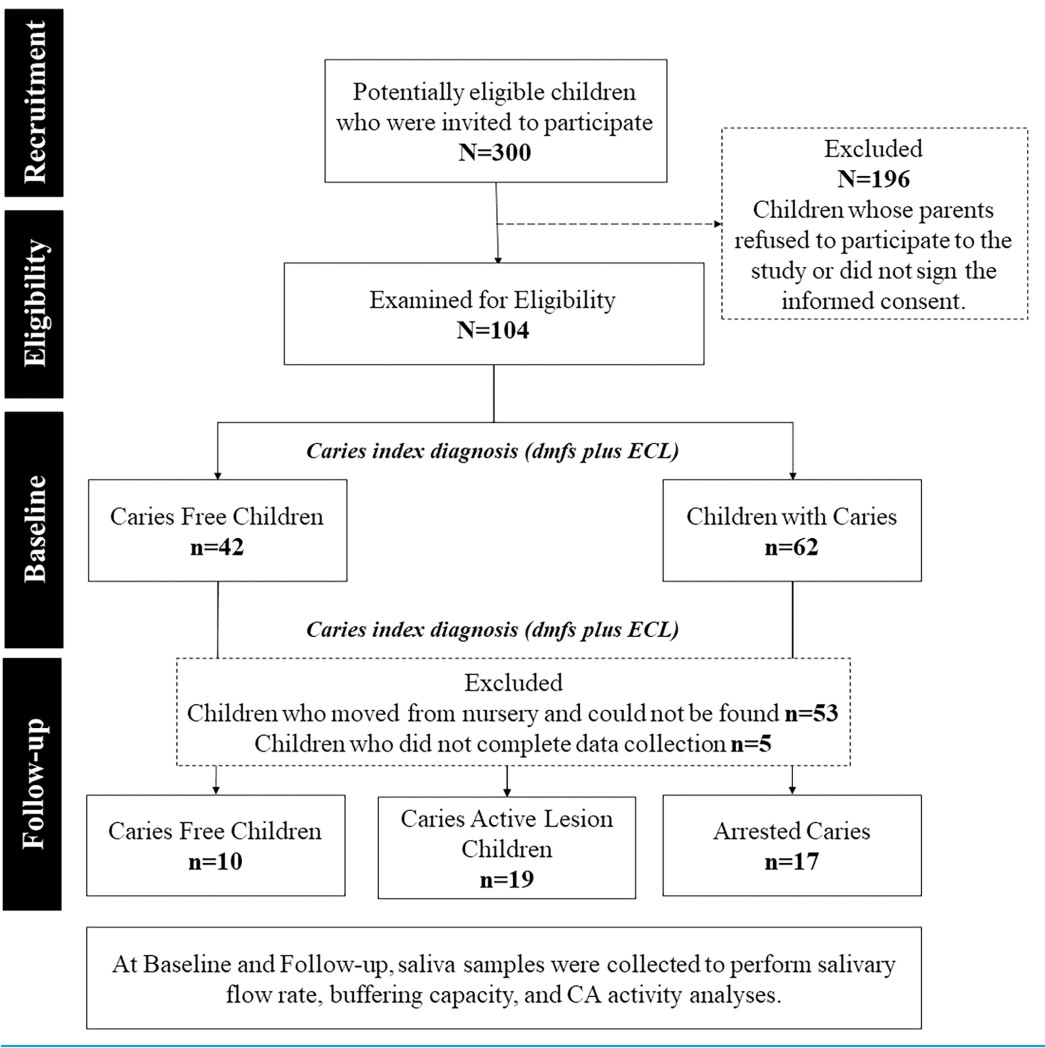

**Figure 1 Flowchart of the research experimental design.** The groups division was done after the study's follow-up period. The comparisons at the baseline were made with the disposition of groups adopted at the end of the study to all comparisons.               

- Caries-free children ($C_{Free}$, $n = 42$): Decayed, missing, and filled surfaces plus early caries lesions (dmft + ECL) = 0.
- Active caries lesion children ($C_{Active}$, $n = 62$). Dmft + ECL ≥ 1.

Children were submitted to visible biofilm examination (*Alaluusua & Malmivirta, 1994*) and saliva collection for salivary flow rate, buffering capacity, and CA VI activity analyses. After a 1-year follow-up, 46 children (0.76 girl to 1 boy) remained in the cohort. The high dropout rate (55.8%) probably occurred since children, who were 6 years old in the first year, moved from their original preschool and could not be found.

Clinical examination and salivary analyses were repeated after 1 year follow-up. According to changes in the caries index, children were assigned into three groups:

- $C_{Free}$, $n = 10$: Children who remained caries-free, dmft + ECL = 0.

- $C_{Active}$, $n = 19$: Children who had developed one or more caries lesions since the beginning of the study, dmft + ECL $\geq$ 1.
- Arrested caries ($C_{Arrested}$, $n = 17$). Children with negative caries increment or who had arrested caries lesions.

## Examiner training, clinical examination, and caries assessment

The adapted version of the World Health Organization diagnostic criteria (*World Health Organization, 2013*) plus the early caries lesions were considered for the clinical examinations (*Assaf et al., 2006*). After calibration, the clinical examinations were performed at baseline and follow-up by only one examiner (first author).

The clinical calibration, using a gold standard for criteria, was held to achieve an acceptable level of agreement before the intraexaminer reliability assessment. First, clinical slides were used to train the examiner regarding the WHO and the ECL criteria and the interexaminer agreement was 80% at baseline and follow up.

Regarding all components of the diagnostic criteria, the intraexaminer reliability was assessed by re-examination of 10% of children, with a one-week interval period and the kappa values at baseline and follow-up for the tooth surfaces were 0.82 and 0.80 respectively. Teeth were cleaned and dried with gauze to favor the identification of early caries lesions.

## Determination of flow rate and buffering capacity

Saliva samples were collected between 9 and 11 a.m. to avoid the influence of circadian rhythm. Children were kept for 2 h without eating, drinking, or chewing gum. The sampling procedure considered the following steps: (1) 5 min to relax, (2) 5 min of stimulated saliva collection, (3) rinse with 5 mL of a 20% sucrose solution for 1 min, and (4) 5 min of stimulated saliva collection.

The stimulus for salivation was reached with Parafilm® (Sigma Chemical Company, St. Louis, MO, USA) chewing. The saliva produced in the initial 30 s was discarded. If the saliva flow rate was low at 5 min, the collection continued for 10 min (*Dawes & Kubieniec, 2004*; *Kirstilä et al., 1998*). After saliva collection, samples were individually kept in labelled closed Falcon® tubes (BD Biosciences, Franklin Lakes, NJ, USA) and transported to the laboratory in a sealed icebox to avoid $CO_2$ loss.

The stimulated salivary flow rate, calculated by measuring the total volume of saliva and dividing it by the collection time, was expressed as mL/min. The Ericsson method was used to determine saliva's buffering capacity (*Ericsson, 1959*). The pH values were assessed using an electronic pH meter (Orion Analyzer Model 420A; Thermo Fisher Scientific Inc., Waltham, MA, USA). After salivary flow rate and buffering capacity analyses, samples were transferred to labelled microtubes, centrifuged at 5,000 rpm for 10 min at 4 °C, stored, and frozen at −40 °C for CA VI activity quantification.

## Quantification of carbonic anhydrase VI

The CA VI activity was determined using zymography (*Kotwica et al., 2006*), modified by *Aidar et al. (2013)*. In short, 10 µL of saliva was added to 10 µL of Tris-buffer (1:1) and

from the final 20 µL, 10 µL and placed in each gel channel of the SDS-PAGE gel (30% T and 0.8% C). Electrophoresis parameters were 1 h: 50 min at 140 V and 4 °C, then, the gel was stained with 0.1% bromothymol blue for 10 min to provide a pH-related color change due to the CA VI reaction with deionized water saturated with $CO_2$, which tends to raise the pH to 7. The luminescence in the bands' area was calculated using the Image J software, providing the CA $VI_{ACT}$ in pixels/area (*Collins, 2007*).

## Statistical analysis

The SPSS package for Windows, version 21.0 (SPSS, Inc., Chicago, IL, USA) and the GraphPad Prism 7.04 program (GraphPad Software for Windows–version 7.04; GraphPad Software, La Jolla, CA, USA) were used for statistical inferences.

Data normality and homogeneity of variances were tested using the Shapiro-Wilk and Levene's tests, respectively. The salivary flow rate and buffering capacity followed the Gaussian distribution. A square root transformation was achieved since CA VI activity data did not follow the Gaussian distribution (*Sámal et al., 1999*). The Box's M test proved the equality of multiple variance-covariance matrices.

The McNemar's test was used to test differences in visible biofilm between baseline and follow-up. The paired t-test was performed to determine whether caries status at baseline and follow-up was significantly different. The salivary flow rate, buffering capacity, and CA VI activity were analyzed using a three-way mixed model analysis of variance to compare the mean differences among groups (caries status) split into within-subjects' factors (annual variation and sucrose rinse). The simple effects test was carried out as a single-step pairwise comparison procedure considering the Bonferroni adjustment applied for multiple comparisons. A 0.05 significance level was established for the analyses.

## RESULTS

The number of caries-affected surfaces increased significantly from 4.26 (SD: 4.77) at the baseline to 5.22 (SD: 6.24) at follow-up, with a significance level of 0.008 alpha level. In children with caries at baseline, the number of affected surfaces was 6.84 (SD: 5.61) for the $C_{Active}$ group and 3.88 (SD: 3.24) for the $C_{Arrested}$ group. The 1-year caries increment for the $C_{Active}$ group was 3.11 (SD: 2.03).

Figure 2 shows the effect of time, rinse, and disease on the salivary flow rate (A), buffering capacity (B), and CA VI activity (C). Material S1 shows more comprehensive statistics.

The salivary flow rate was significantly reciprocally influenced by time, caries disease, and sucrose rinse. At baseline, salivary flow rate significantly increased after sucrose rinse in the three investigated groups, whereas, at follow-up, this increase was observed only in the arrested caries group. Furthermore, results showed a significantly increase in salivary flow rate at follow-up for the three groups in the pre-rinse situation and for the $C_{Arrested}$ group in the post-rinse situation. Although the effect of caries disease was codependent on time and rinse ($\alpha = 0.03$, $\beta - 1 = 0.66$, $\eta p^2 = 0.151$), simple effects revealed that this effect was significant at follow-up in the post-rinse condition ($\alpha = 0.028$, $\beta - 1 = 0.67$, $\eta p^2 = 0.15$).

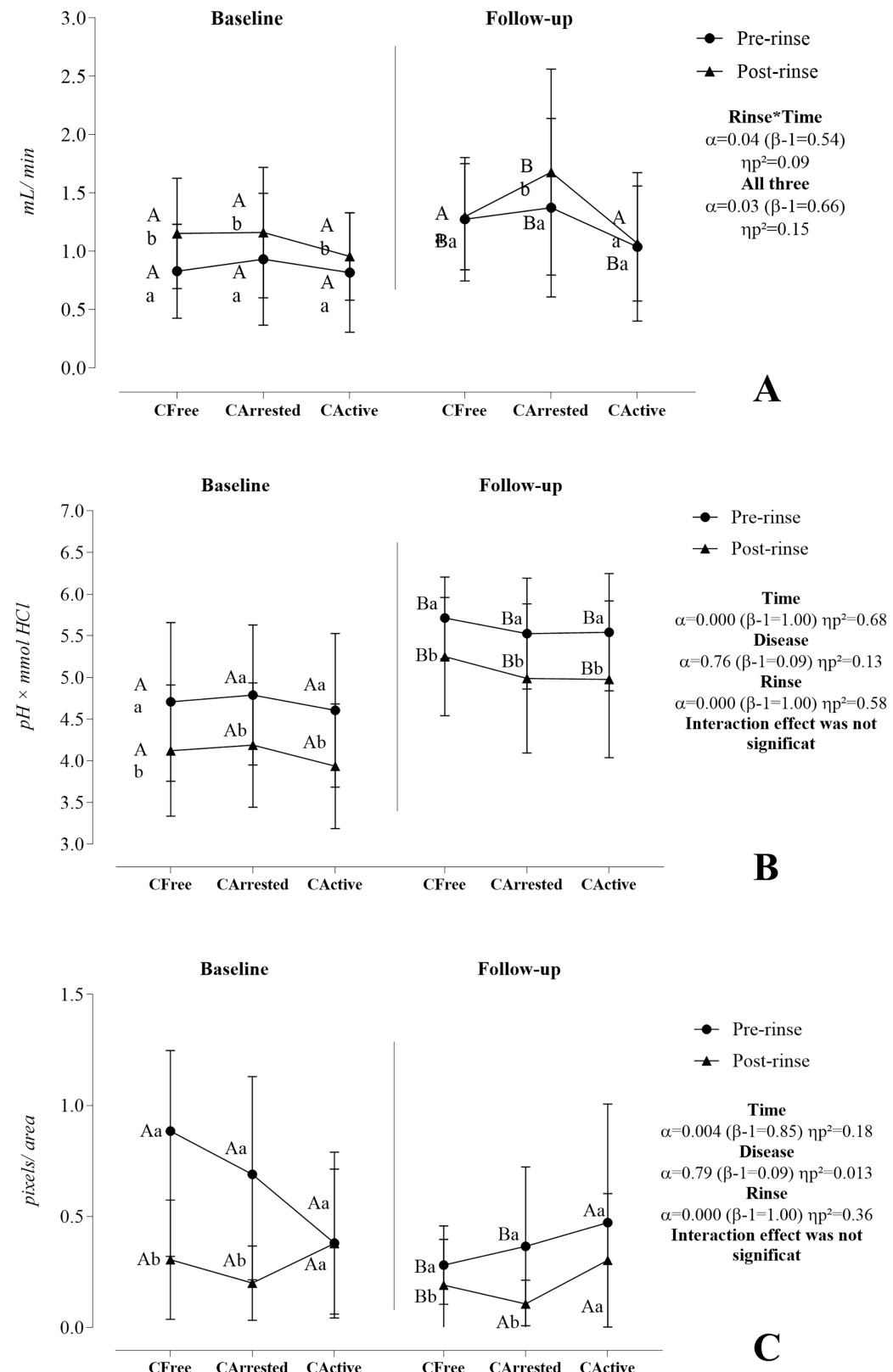

**Figure 2 Effect of time, rinse, and disease on the salivary flow rate (A), buffering capacity (B), and CA VI activity (C).** A three-way mixed model analysis of variance (ANOVA) was conducted on a sample
Time and sucrose rinse independently influenced the buffering capacity. A reduction in BC was observed after sucrose rinse, but it tended to be similar between groups at baseline (6.27 [SD 0.69]; $p = 0.937$) and at follow-up (0.54 [SD 0.58]; $p = 0.906$). A significant improvement in the buffering capacity was found at follow-up compared with baseline at pre-rinse and post-rinse.

Time and sucrose rinse significantly affected the CA VI activity but did not exhibit a significant interaction. Over time, the CA VI activity significantly decreased in the $C_{Free}$ and $C_{Arrested}$ groups in pre-rinse situation. However, the behavior of CA VI throughout time did not change in the $C_{Active}$ group. At baseline and follow-up, CA VI activity significantly decreased after sucrose rinse in $C_{Free}$ and $C_{Arrested}$ groups but not in the $C_{Active}$ group. In addition, we could not find any significant difference between groups concerning the CA VI activity at baseline and follow-up.

## DISCUSSION

This study showed that the CA VI activity reduced over time and after exposure to sucrose rinse whether children were caries-free or had only arrested caries. The CA VI activity remained stable over time in children with active caries lesions, even after the cariogenic challenge. Controversially, the reduction in the CA VI activity at the follow-up was accompanied by an expressive increase in the buffering capacity in saliva.

The $C_{Free}$ and $C_{Arrested}$ groups showed significant decreases in the CA VI activity after sucrose rinse and at follow-up. The adaptive response of the isoenzyme to a medium of high acid production may reduce its activity due to an inhibitory effect mediated by sucrose. Similarly, a previous study found a pronounced decrease in the initial activity of CA VI activity after sucrose rinse when the mechanical control of biofilm was discontinued (*Lima-Holanda et al., 2021*). These authors argued that poor oral hygiene causes a high and sustained concentration of metabolites, which could be responsible for a change in CA VI enzymatic activity.

The $C_{Active}$ group did not exhibit significant changes in CA VI activity following the sucrose rinse and at follow-up. One plausible explanation could be the frequent exposure to cariogenic challenges experienced by this particular group (*Nobre dos Santos et al., 2002*;

*Parisotto et al., 2010*). Frequent pH drop events could be responsible for the reduced source of $HCO_3^-$ due to more significant amounts of acid production by microbial metabolism during famine events (*Takahashi, 2015*). Consequently, the enzyme could no longer change its activity after a sucrose rinse. *Kazokaitė et al. (2015)* strengthened our assumption by showing that CA VI is more stable in slightly acidic conditions than in neutral ones.

The hypothesis here is that the dysbiotic events ruled by caries promote an enzymatic adaptation for more intense chemical aggression. Notably, caries undergoes a biochemical shift toward disease (*Marsh, 2018*) and structural and conformational changes in salivary molecules (*Zeng, 2011*; *Buonanno et al., 2018*). Thus, investigating the ecological interactions among microbiome, electrolytes, and glycolytic metabolites and their respective impact on the phenotypic structure of CA VI activity can be an exciting topic for future research.

Expanding the scope of this discussion, could the transient enzymatic changes in CA VI activity affect the acid-base control in the oral cavity? Addressing this question, we noticed a significant decrease in buffering capacity at baseline and follow-up after sucrose rinse. Sugar metabolization should consume the acid of the saliva buffers, making the saliva medium less resistant to pH changes. However, our study design was insufficient to identify the effect of sucrose-mediated enzymatic changes on the buffer capacity regarding catalytical efficiency.

Caries disease did not seem to impact CA VI activity, unlike other studies (*Frasseto et al., 2012*; *Picco et al., 2017*, *2022*; *Lima-Holanda et al., 2021*; *de Sousa et al., 2021*; *de-Sousa, Lima-Holanda & Nobre-Dos-Santos, 2021*). These authors performed cross-sectional studies and found a higher CA VI activity in the saliva of children having caries. This conflicting outcome may be due to considerable inter-individual variation and to the small sample size of our study, which was demonstrated by the low power of the analysis ($\beta - 1 = 0.09$). Buffering capacity results showed the same trend and may be explained similarly.

As expected, our study also showed that at baseline, the saliva flow rate increased after sucrose rinses in three groups with no statistical difference between them, which could be promoted by the mechanical and gustatory stimulation reached during the sucrose rinse and parafilm chewing (*Proctor, 2016*). In this case, the lack of statistical difference can be interpreted as a consequence of the healthy salivary flow rates. Preliminary data have shown that the relationship between saliva flow rate and dental caries has no predictive value for disease occurrence when the flow is within physiological range (*Lenander-Lumikari & Loimaranta, 2000*). Note that an increase in the saliva flow rate was perceived at the follow-up. We speculate that the maturation of salivary function may be the leading cause of this outcome. The rise in salivary flow could explain the independent increase in buffering capacity regardless of the caries index after the follow-up period.

We must point out some limitations of this research. Firstly, the high sample dropout compromised the power of the analyses regarding the difference between groups. Secondly, the analysis was restricted to saliva samples. Thus, we could not infer the impact of the CA VI function in the acid-base equilibrium at the biofilm-tooth interface. Thirdly,

the cariogenic challenge with a 20% sucrose solution gives limited information concerning the clinical reality, mainly due to the complexity of human diet and its dependence on other carbohydrates that can impair the physicochemical properties of saliva and enzymatic behavior of iso-enzymes such as CA VI. Finally, although this study included the early caries lesion criteria which increases the dmfs/dmft indexes, bitewing radiographs were not used, thus some lesions at the distal surface of the first primary molars may have been underdiagnosed.

## CONCLUSIONS

In conclusion, the study highlights two distinct mechanisms: one linked to caries activity and the other to the salivary function maturation. The sucrose rinse lowered buffering capacity regardless of caries status, while caries activity may disrupt CA VI activity response, affecting enzyme stimulation during a cariogenic challenge. After a 1-year follow-up, the rise in the salivary flow might enhance buffering capacity as a chemical shift toward a healthier state. Yet, this improvement may not be mirrored in enzyme activity among caries-active children.

## ACKNOWLEDGEMENTS

This article was based on a thesis submitted by the first author to the Piracicaba Dental School, Universidade Estadual de Campinas, in partial fulfillment of the requirements for a DDS degree in Dentistry (Pediatric Dentistry area). We especially thank the volunteers and their parents for participating in this research.

### Funding

This work was supported by the São Paulo Research Foundation, Grants No 2012/02516-1 and 2012/15834-1. The funders had no role in study design, data collection and analysis, decision to publish, or preparation of the manuscript.

### Grant Disclosures

The following grant information was disclosed by the authors:
São Paulo Research Foundation: 2012/02516-1, 2012/15834-1.

### Competing Interests

The authors declare that they have no competing interests.

### Author Contributions

- Thayse Rodrigues de Souza conceived and designed the experiments, performed the experiments, analyzed the data, prepared figures and/or tables, authored or reviewed drafts of the article, and approved the final draft.
- Bruna Raquel Zancope performed the experiments, prepared figures and/or tables, and approved the final draft.
- Emerson Tavares de Sousa analyzed the data, prepared figures and/or tables, authored or reviewed drafts of the article, and approved the final draft.
- Thais Manzano Parisotto analyzed the data, authored or reviewed drafts of the article, and approved the final draft.
- Marcelo Rocha Marques performed the experiments, authored or reviewed drafts of the article, and approved the final draft.
- Marinês Nobre dos Santos conceived and designed the experiments, analyzed the data, authored or reviewed drafts of the article, and approved the final draft.

### Human Ethics

The following information was supplied relating to ethical approvals (*i.e.*, approving body and any reference numbers):

The Ethics Committee in Research of the Piracicaba Dental School, University of Campinas, granted Ethical approval to carry out the study within its facilities (Ethical Application Ref: 0142012).

### Data Availability

The raw data and statistical analysis are available in the Supplemental Files.

### Supplemental Information

Supplemental information for this article can be found online at http://dx.doi.org/10.7717/peerj.17429#supplemental-information.

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
