# Peer review of "Sucrose rinse modulates the salivary behavior of carbonic anhydrase VI and its buffering capacity: a longitudinal study in 4 to 6.5-year-old children"

_PeerJ, doi:10.7717/peerj.17429_

## Round 0.1 · original submission · Major Revisions

Dear Authors,

Both reviewers raised several serious concerns regarding the study methodology and reporting. Please resubmit the manuscript only after extensive revision, carefully addressing all issues of concern. If you are not willing to do so, I would suggest you withdraw it and submit it elsewhere.

With kind regards,
Nikolaos Gkantidis
Academic Editor

**Language Note:** The review process has identified that the English language must be improved. PeerJ can provide language editing services - please contact us at [email protected] for pricing (be sure to provide your manuscript number and title). Alternatively, you should make your own arrangements to improve the language quality and provide details in your response letter. – PeerJ Staff

Reviewer 1 ·

Basic reporting

1- It is mentioned that there are 2 tables in the beginning of the manuscript where I found only one.
2- In general, the manuscript was poorly written.
3- Background or abstract was repeated.
4- Please provide abbreviation list to make it easy to follow.
5- Lines 51 & 249: HCO3- or HCO3-
6- In many places in the manuscript the abbreviations were not correct e.g. lines 34, 56 & 64: CA VI not AC VI
7- Provide reference for the sentence in line 58-60 (Some of these attempts have suggested that CA VI would protect teeth by catalyzing the removal of microbe-………)
8- Some unclear sentences e.g., line 68-69 (behavior of AC VI rinse?)
9- References in the refences are not listed correctly. E.g., PMID was mention in the end of some references.
10- Again: the language should be improved. In line 239 you write reduce your activity.
11- Again: Line 223 AC group not CA group.
12- Provide the reference number for the ethical approval.
13- In Table one either to use arrested caries or caries arrested in the whole article. Constancy is very important for reader in order to follow the article.
14- Line 225-226 no differences in what? Please clarify the sentence.

Experimental design

1- It was mentioned in several places that the number of children is 47. Shouldn’t it be 46 (10+17+19= 46)? Will this have an affect on the results since it is already a small sample size?
2- It was mentioned in line 96 that (The calculated number was 35 children for each group) and you end up with less than 35 (10, 17, 19) due to many reasons. I am no sure that you can have a determined conclusion in such a small sample size/number specially for the controls or CF group of participants. What do you think?
3- The results and hypothesis should focus when writing about the longitudinal study not only after 20% sucrose rinse as it is clearly mentioned in the title.
4- For biofilm examination: Was the method of examination the same for baseline and follow up?
5- More details should be mentioned in the method section or should be referred to a previous article. For example, were the participants or their parents informed what to do before the biofilm examination.
6- In the method section, the name of companies for materials and software should be included.
7- There were no results in relation to biofilm examination. Why is it mentioned in the material and method section?

Validity of the findings

1- There is no figure ligand in Figure 2. The statistical test was not described.
2- It is not clear in Figure 2, what are the letters a, b, A and B for. In my opinion (I might be wrong) there is a mistake in Figure 2. C regarding the capital and small letter e.g. in AC group it should be Bb not Ab. This should be clear for the readers.
3- Line 224-225, you mention that there is a significant decrease in the CA VI activity in the follow up category in caries free group. Is that correct

Reviewer 2 ·

Basic reporting

This is an interesting study on the possible effect of sucrose rinsing to salivary qualities. The manuscript follows the professional structure of a scientific article, and the raw data was shared. However, the following issues need to be addressed.
a) The manuscript needs language editing, e.g., avoid too many abbreviations that make difficult to understand the text. In lines 64 and 69, the abbreviation “AC VI” seems to be wrong. It is unclear what the authors mean in lines 59-61.
b) The headings and labels of tables and figures need improvement. For instance, what is (A:P) in table 1? What are “Aa”, “Ab”, “Ba”, and “Bb” in figure 2?

Experimental design

The authors “attempted to decipher how CA VIACT could influence the salivary
buffering capacity over time in children at risk for dental caries development.” (lines 73-74). In the next lines (75-76), they try to further specify the aim by writing “Thus, this cohort study investigated how salivary carbonic anhydrase VI activity and buffering capacity behave after a 20% sucrose rinse in 4 to 6.5 years-old children”. It is obvious that the text in the latter lines describes a different aim than in the former. I suggest the removal of lines 73-74.

The authors do not provide the scientific background and rationale that support their hypothesis, i.e., one mouth rinse with 5 ml of a 20% sucrose solution alters the physiology of salivary function (flow rate, buffering capacity and carbonic anhydrase activity).

Several points in the study design also need to be clarified.
a) How many examiners were involved, and how often were they calibrated during the study period? Were the same examiners who examined the children on the two occasions (baseline and follow-up)?
b) Caries detection by only clinical examination is not sufficient to determine the caries activity status of a patient. X-ray imaging is necessary. The small number of participants in the groups of the study makes this requirement imperative. It must be discussed as a limitation.
c) The purpose for doing this very coarse registration of dental plaque on the buccal surface of the upper incisors is unclear. Since it gave no information relevant to the aim, it is better to remove.
d) The flow rate of stimulated saliva was not determined in the same way for all participants. In some of them the collection time was increased because, they produced low volume of saliva during the first 5 min. How low volume did you consider insufficient? How many of the participants needed more time?
e) The flow rate and the composition of saliva change as the collection time increases. Did the authors take it into consideration?
f) Sucrose rinsing causes gustatory stimulation of saliva secretion on top of the mechanical due to the parafilm chewing. Moreover, rinsing the mouth with 5 ml sucrose solution immediately before the saliva sampling dilutes the saliva in the mouth. To avoid escape of carbon dioxide, saliva must be sampled in a closed device and not the way it was done in this paper. Transporting in a closed box has no effect on the gas escape. Did the authors consider all these methodological issues?
g) It is unclear how the enzymatic activity of carbonic anhydrase was measured. Did the authors followed the colour change of the pH indicator in time? Did the authors relate the activity to the amount of protein in the CA band of the gel? Is it possible to have variations in the activity per mg protein?
h) The reference “Ericsson & Hardwick, 1978” for the determination of the buffering capacity of saliva (line 160) is not correct. Probably you mean “Ericsson, 1959”.

Validity of the findings

No more comments

Additional comments

Undoubtfully, the results are peculiar, unexpected and need explanation. The discussion is just for this reason. The authors have to interpret their finding with rational explanations. Instead, the present discussion contains new hypothetical aspects with no support by the results.

---

## Round 0.2 · accepted · Accept

All reviewers' concerns have been adequately addressed and I can now confirm that your manuscript can be accepted for publication in its present form.

Reviewer 2 ·

Basic reporting

No comment

Experimental design

No comment

Validity of the findings

No comment